# Atg2 Regulates Cellular and Humoral Immunity in *Drosophila*

**DOI:** 10.3390/insects14080706

**Published:** 2023-08-14

**Authors:** Bo Qin, Shichao Yu, Qiming Chen, Li Hua Jin

**Affiliations:** College of Life Science, Northeast Forestry University, Harbin 150040, China; qin121bo@163.com (B.Q.); yushichao1990@hotmail.com (S.Y.); chenqiming237@163.com (Q.C.)

**Keywords:** autophagy, Atg2, innate immunity, *Drosophila*

## Abstract

**Simple Summary:**

In *Drosophila melanogaster*, autophagy regulates development and stress responses. However, the mechanisms underlying how autophagy modulates innate immunity still need further investigation. In this study, we identified Atg2, an autophagy-related protein, as playing an important role in controlling innate immunity. We revealed that inhibiting *Atg2* induced melanotic nodule formation, disrupted phagocytosis in vivo, altered the expression of AMP-encoding genes and impaired the ability to resist bacterial infections. This study will advance the understanding of the relationship between autophagy and innate immunity.

**Abstract:**

Autophagy is a process that promotes the lysosomal degradation of cytoplasmic proteins and is highly conserved in eukaryotic organisms. Autophagy maintains homeostasis in organisms and regulates multiple developmental processes, and autophagy disruption is related to human diseases. However, the functional roles of autophagy in mediating innate immune responses are largely unknown. In this study, we sought to understand how *Atg2*, an autophagy-related gene, functions in the innate immunity of *Drosophila melanogaster*. The results showed that a large number of melanotic nodules were produced upon inhibition of *Atg2*. In addition, inhibiting *Atg2* suppressed the phagocytosis of latex beads, *Staphylococcus aureus* and *Escherichia coli*; the proportion of Nimrod C1 (one of the phagocytosis receptors)-positive hemocytes also decreased. Moreover, inhibiting *Atg2* altered actin cytoskeleton patterns, showing longer filopodia but with decreased numbers of filopodia. The expression of AMP-encoding genes was altered by inhibiting *Atg2*. *Drosomycin* was upregulated, and the transcript levels of *Attacin-A, Diptericin* and *Metchnikowin* were decreased. Finally, the above alterations caused by the inhibition of *Atg2* prevented flies from resisting invading pathogens, showing that flies with low expression of *Atg2* were highly susceptible to *Staphylococcus aureus* and *Erwinia carotovora carotovora 15* infections. In conclusion, Atg2 regulated both cellular and humoral innate immunity in *Drosophila*. We have identified Atg2 as a crucial regulator in mediating the homeostasis of immunity, which further established the interactions between autophagy and innate immunity.

## 1. Introduction

Despite the lack of adaptive immunity in *Drosophila melanogaster*, its innate immune system, acting as the first line of defense, shares similarities with mammals both in physiology and signaling pathways [1]. Thus, in recent decades, *Drosophila* has emerged as an ideal model for studying innate immunity [2]. The innate immune response of *Drosophila* is mainly composed of two parts: the cellular immune response and the humoral immune response [1,2]. The cellular immune response is executed by three types of hemocytes (blood cells of *Drosophila*), in which plasmatocytes constitute 95% of the total hemocytes and play an important role in phagocytosis [3,4,5]. Macrophage-like plasmatocytes are capable of engulfing pathogens and apoptotic debris produced during development [3,4,5]; the dynamics of the actin cytoskeleton and multiple phagocytosis receptors, including Nimrod C1 (NimC1), Eater, Draper, Croquemort, Down syndrome cell adhesion molecule 1 (Dscam1) and class C scavenger receptor I (dSr-CI), are required in this process [6,7,8]. Crystal cells, another type of hemocyte, can heal wounds via melanization responses [1,2]. In addition, although rare in healthy larvae, lamellocytes are largely induced upon immune challenge and are in charge of encapsulating large foreign bodies that cannot be phagocytosed [9,10]. As another aspect of innate immunity, the humoral immune response largely depends on antimicrobial peptides (AMPs). When encountered with immune challenge, fat bodies will secrete AMPs into the hemolymph to combat invading pathogens [1,2]. In *Drosophila*, the induction of AMPs is mainly regulated by two NF-κB-related signaling pathways, Toll and Immune deficiency (Imd). Moreover, the Toll pathway shows higher affinity for fungi and Gram-positive bacteria, whereas the Imd pathway is more likely to be involved in the response to Gram-negative bacteria [1,2].

Autophagy is a conserved process that promotes the intracellular degradation of cytoplasmic proteins or organelles with lysosomes in eukaryotic organisms [11,12]. Under normal conditions, the activity of autophagy is maintained at a relatively low level [13]. However, upon starvation or other stresses, autophagy is strongly induced [14,15]. Thus, autophagy plays a critical role in maintaining the homeostasis of cells and organisms, and autophagy has been shown to be versatile in regulating development and the stress response [16,17,18]. In addition, autophagy dysfunction is associated with human diseases, including neurodegeneration and cancer [19]. In *Drosophila*, autophagy activity is modulated by a series of *autophagy-related* (*Atg*) genes, which are highly conserved from yeast to mammals [20,21]. The biological functions of some Atg proteins are relatively well characterized; for instance, Atg1 regulates cell growth and apoptotic cell death and maintains germline stem cells by modulating mitochondrial dynamics [22,23]. Inhibiting *Atg5* induces developmental delay and ataxia, while Atg8 is involved in larval midgut cell programmed cell death and determination of adult lifespan [24,25,26]. However, the functional role of Atg2 is largely unknown. Moreover, although autophagy was shown to restrict bacterial infection and viral replication, the detailed mechanism still warrants further investigation.

In this study, we focused on Atg2, an Atg protein with a lesser-known biological function, and its role in regulating innate immunity. We showed that inhibiting *Atg2* in hemocytes induced the formation of melanotic nodules and disrupted phagocytosis. In addition, the proportion of NimC1-positive hemocytes decreased, and hemocytes displayed altered actin cytoskeleton distribution upon inhibition of *Atg2*. Flies with low *Atg2* expression showed altered AMP-encoding gene expression and succumbed to bacterial infection. In summary, we identified *Atg2* as a crucial regulator of innate immunity, which established the connection between autophagy and innate immune responses.

## 2. Materials and Methods

### 2.1. Drosophila Strains and Culture Conditions

*w^1118^* (BL5905, used as the wild type), *Atg2^EP3697^/TM6B*, *Tb* (BL17156) and *Cg-Gal4* (BL7011) were obtained from the Bloomington *Drosophila* Stock Center (BDSC). *UAS-Atg2^RNAi^* (THU3698) was obtained from the TsingHua Fly Center. *UAS-Imd^RNAi^* (V9253) were obtained from the Vienna *Drosophila* Resource Center. *Pxn-Gal4*; *UAS-GFP* was a gift from Professor Norbert Perrimon. *UAS-Dif^RNAi^* was a gift from Professor Lei Xue. The flies were fed standard cornmeal-yeast medium and maintained at 25 °C in an incubator. To activate the UAS/Gal4 system, the conditions were shifted to 29 °C for at least 48 h.

### 2.2. In Vivo Phagocytosis

Ten to fifteen wandering 3rd instar larvae from each group were injected with 200 nl of latex beads (10%, F8821 and F8823, Invitrogen, Waltham, MA, USA), pHrodo-*Escherichia coli* (*E. coli*) (1 mg/mL, P35361, Thermo Fisher Scientific, Waltham, MA, USA) or pHrodo-*Staphylococcus aureus* (*S. aureus*) (1 mg/mL, A10010, Thermo Scientific) using an injector (PICOSPRITZER III, Parker Hannifin, Mayfield Heights, OH, USA). Next, injected larvae were placed in a humid chamber at room temperature for 1 h. To obtain circulating hemocytes, the larval cuticle was ripped gently with fine forceps (Dumont, Montignez, Switzerland) when submerging larvae in 10 μL of phosphate-buffered saline (PBS) and then incubated in a dark chamber at room temperature for 30 min on an adhesive glass slide. After fixation with 4% paraformaldehyde (PFA) for 10 min and several careful washes with PBS containing 0.1% Tween 20 (PBST), hemocytes were mounted with 70% glycerol and observed under a fluorescence microscope (Axio Scope A1, Zeiss, Germany). The images were captured randomly to avoid bias. To calculate the average phagocytosed particles per hemocyte (phagocytic index), the total number of particles (or the total fluorescence intensities) and the number of hemocytes of each image were determined with ImageJ software. Phagocytic index = Total number of particles (or total fluorescence intensities)/The total number of hemocytes. Engulfing cell (%) = The number of hemocytes with phagocytosed particles/The total number of hemocytes. Eight larvae from each group were employed in the bleeding. For each group, at least 500 hemocytes were analyzed.

### 2.3. Immunostaining of Melanotic Nodules

Melanotic nodules were directly dissected from wandering 3rd instar larvae and fixed with 4% PFA for 30 min. Following 3 washes (5 min/wash) with PBST, melanotic nodules were blocked with PBST containing 5% goat serum for 30 min and incubated with anti-L1 antibodies (a gift from Professor István Andó) at 4 °C overnight. After several rinses in PBST, melanotic nodules were incubated with Alexa Fluor 568-conjugated secondary antibodies (A-21124, Thermo Fisher Scientific). Following washing with PBST 3 times, dissected tissues were stained with Hoechst dye (H1399, Invitrogen) to visualize cell nuclei, mounted with Slowfade (S36967, Invitrogen), and observed under a microscope (Axio Scope A1, Zeiss, Germany). For each group, at least 15 larvae were analyzed.

### 2.4. Hemocyte Assays

P1 staining: eight wandering 3rd instar larvae from each group were bled in 10 μL of PBS, and then hemocytes were transferred to an adhesive glass slide. After incubation at room temperature for 30 min, hemocytes were fixed with 4% PFA, blocked with PBST containing 5% goat serum and incubated with anti-P1 (a gift from Professor István Andó) at 4 °C overnight. The next day, hemocytes were incubated with Alexa Fluor 568-conjugated secondary antibodies (A-11004, Invitrogen) and Hoechst dye for 2 h and 10 min, respectively, and mounted with Slowfade.

F-actin staining: eight wandering 3rd instar larvae were bled in 10 μL of PBS, incubated at room temperature for 30 min, and fixed with 4% PFA for 10 min. After 3 washes, hemocytes were incubated with Alexa Fluor 568-conjugated phalloidin (A12380, Invitrogen) and Hoechst dye for 30 min and 10 min, respectively. Then, hemocytes were mounted with Slowfade.

TUNEL and 7-AAD staining: To examine cell apoptosis and death, hemocytes were incubated with a TUNEL kit (11684795910, Roche, Basel, Switzerland) and 7-AAD (5 μg/mL, A1310, Invitrogen), respectively, according to the manufacturer’s instructions.

All stained sections were analyzed under a fluorescence microscope (Axio Scope A1, Zeiss, Germany).

### 2.5. Quantitative Real-Time PCR (qRT—PCR)

Eight adult flies from each group were collected in TRIzol reagent (Invitrogen) and used to isolate total RNA. After synthesizing cDNA using reverse transcriptase (Promega), qRT—PCR was performed with SYBR Green I Master Mix (Roche) on a Roche LightCycler 480 real-time PCR system. The transcript levels of AMP-encoding genes were normalized to *ribosomal protein 49* (*rp49*), the primer sequences used are shown in Table 1. The fold change was calculated with the 2^−∆∆Ct^ method [27]. The experiments were executed with three independent replications.

### 2.6. Survival Rates upon Bacterial Infection

Three- to five-day-old adult flies (15 males and 15 females/group) were injected with 50 nl of *S. aureus* (OD = 0.0005) or *Erwinia carotovora carotovora 15* (*Ecc-15*, OD = 0.4). After injection, flies were incubated at 29 °C, and surviving flies were counted every 24 h. In addition, injected flies were shifted to a new vial every 48 h. The experiment was repeated independently three times.

### 2.7. Statistical Analysis

All microscopic images were analyzed with ImageJ software. The statistical analysis was performed with GraphPad Prism 6.0, and *p* values were determined with unpaired Student’s t test or log-rank (Mantel—Cox) test (survival assay). The thresholds for statistical significance were * *p* < 0.05, ** *p* < 0.01, *** *p* < 0.001 and **** *p* < 0.0001. All experiments were repeated independently at least three times. The error bars in all column diagrams indicate the means ± SDs.

## 3. Results

### 3.1. Inhibiting Atg2 Induced Massive Melanotic Nodule Formation

To examine the functional role of Atg2 in the innate immunity of *Drosophila*, we first knocked down *Atg2* in the blood system with the hemocyte-specific *Gal4* driver *Pxn-Gal4; UAS-GFP.* Surprisingly, massive melanotic nodules emerged in *Pxn* > *GFP/Atg2^RNAi^* larvae, while no melanotic nodules were observed in the control group (Figure 1A,B). To confirm this result, we utilized *Cg*-*Gal4*, an immune system-specific driver (hemocytes and fat bodies), to suppress *Atg2* and observed a large number of melanotic nodules in *Cg* > *Atg2^RNAi^* larvae (Figure 1C,D). Consistent with the above data, loss of *Atg2* by using mutant *Atg2^EP3697^/Tb* induced melanotic nodule formation (Figure 1E,F). Moreover, after immunostaining with anti-L1 antibodies that label lamellocytes, melanotic nodules showed GFP-positive (signals from *Pxn-Gal4*; *UAS-GFP*) and L1-positive signals, indicating that plasmatocytes and lamellocytes were deposited on the melanotic nodules (Figure 1G–K). To exclude non-specific fluorescence signals during the immunostaining, we stained melanotic nodules without using anti-L1 antibodies but using Alexa Fluor-conjugated secondary antibodies; no red fluorescence was observed, indicating that the fluorescence signals were specific (Appendix A). However, TUNEL assay and 7-AAD staining showed that no significant cell apoptosis or cell death occurred in circulating hemocytes of *Pxn* > *GFP/Atg2^RNAi^* larvae (Appendix A). Collectively, inhibition of *Atg2* significantly induced melanotic nodule formation.

### 3.2. Inhibiting Atg2 Impaired In Vivo Phagocytosis

Next, we sought to determine the effects of inhibiting *Atg2* on hemocyte function by examining in vivo phagocytosis. First, latex beads, nonpathogenic particles with red fluorescence, were incorporated, and the results showed that compared with the control group, the proportion of hemocytes with phagocytosed beads (engulfing hemocytes) significantly decreased in *Pxn* > *GFP/Atg2^RNAi^* hemocytes; the average number of phagocytosed beads per hemocyte (phagocytic index) also decreased (Figure 2A–D). In addition, the phagocytosis of the Gram-positive bacterium *S. aureus* was disrupted upon inhibition of *Atg2* (Figure 2E–H). Given that different bacteria might activate different phagocytosis receptors, the Gram-negative bacterium *E. coli* was utilized. Similarly, the proportion of hemocytes with phagocytosed pHrodo-*E.coli* and the phagocytic index decreased in *Pxn* > *GFP/Atg2^RNAi^*, showing that in vivo phagocytosis was disrupted by inhibiting *Atg2* (Figure 2I–L). Moreover, these results were confirmed in *Cg* > *Atg2^RNAi^* and *Atg2^EP3697^/Tb* larvae (Appendix A), indicating that Atg2 facilitates in vivo phagocytosis.

### 3.3. Inhibiting Atg2 Altered the Proportion of NimC1-Positive Hemocytes and the Actin Cytoskeleton

To further explore the mechanism of impaired phagocytosis upon inhibition of *Atg2*, we examined the expression of NimC1, a phagocytosis receptor and a marker for mature plasmatocytes, with anti-P1 antibodies. A total of 95.14% of the hemocytes from the control group were P1-positive, while only 41.16% of hemocytes were labeled with P1 in *Pxn* > *GFP/Atg2^RNAi^* hemocytes, indicating that inhibiting *Atg2* decreased the proportion of NimC1-positive hemocytes (Figure 3A,B); this was also observed in *Cg* > *Atg2^RNAi^* and *Atg2^EP3697^/Tb* hemocytes upon latex bead injection (Figure 3D–F,H–J). In addition, we compared the phagocytosis capacity between P1-positive and P1-negative hemocytes and showed that P1-positive hemocytes significantly phagocytosed more latex beads (Figure 3G,K). We stained hemocytes from *w^1118^* larvae (with or without being injected with latex beads) only with Alexa Fluor 568-conjugated secondary antibodies, and no red fluorescence was observed (Appendix A); this result suggested that the red fluorescence in Figure 3 was specific.

Next, we labeled the F-actin of hemocytes with phalloidin to examine whether inhibiting *Atg2* affected the actin cytoskeleton. Although the average length of filopodia was longer in *Pxn* > *GFP/Atg2^RNAi^* hemocytes than in the control group, the average number of filopodia decreased (Figure 4A–D); this result suggested that hemocytes displayed aberrant F-actin distribution. We confirmed this result in *Cg* > *Atg2^RNAi^* and *Atg2^EP3697^/Tb* larvae (Figure 4E–L). In summary, Atg2 plays an important role in maintaining normal proportion of NimC1-positive hemocytes and the actin cytoskeleton.

### 3.4. Inhibiting Atg2 Altered AMP-Encoding Gene Expression

The above data showed that Atg2 participated in cellular immune responses. Thus, we sought to determine whether Atg2 also functions in humoral immune responses, another arm of *Drosophila* innate immunity. To verify this, we knocked down *Atg2* in the immune system with *Cg-Gal4* and checked the expression of a series of AMP-encoding genes, including *Drosomycin* (*Drs*), *Attacin-A* (*AttA*), *Diptericin* (*Dpt*)*, Metchnikowin* (*Mtk*) and *Defensin* (*Def*) in larvae. Compared with the control group, the transcript level of *Drs* increased in *Cg* > *Atg2^RNAi^* larvae, whereas the expressions of *Atta*, *Dpt* and *Mtk* significantly decreased (Figure 5). This result showed that Atg2 had a role in maintaining the homeostasis of AMP production.

### 3.5. Flies Were Susceptible to Bacterial Infection upon Inhibition of Atg2

Finally, to examine whether altered innate immune responses caused by inhibiting *Atg2* would affect the ability to resist infection, adult flies were septic-infected with *S. aureus*, and survival rates were determined; flies with decreased expression of Dif, a key regulator of the Toll pathway, were employed as the positive control. After infection for 7 days, 40.00% of control flies survived, while the survival rate decreased significantly in *Pxn* > *GFP/Atg2^RNAi^* and *Pxn* > *GFP/Dif^RNAi^* flies, showing that only 8.31 and 1.90% flies survived, respectively (Figure 6A). Consistent with this, *Cg* > *Atg2^RNAi^* and *Cg* > *Dif^RNAi^* flies succumbed to *S. aureus* infection (Figure 6B). In addition, *Ecc-15* was incorporated to infect flies, and flies upon inhibition of *Imd* were used as the positive control. Similar to *S. aureus* infection, flies showed significantly lower survival rates in the *Pxn* > *GFP/Atg2^RNAi^*, *Pxn* > *GFP/Imd^RNAi^*, *Cg* > *Atg2^RNAi^* and *Cg* > *Imd^RNAi^* groups (Figure 6C,D). Collectively, these results indicated that Atg2 had a critical role in resisting bacterial infection.

## 4. Discussion

Autophagy is a conserved and crucial intracellular process that maintains the homeostasis of organisms. Several studies have reported the role of autophagy in regulating immunity [28,29,30]. Upon binding to the peptidoglycan recognition receptor PGRP-LE, the peptidoglycan of *Listeria monocytogenes* (*L. monocytogenes*) induces autophagy, and the bacteria are eliminated through Atg8 [28]. The clearance of *Mycobacterium marinum* and *Salmonella enterica* also involves Atg8 [28]. In addition, autophagy was shown to play an important role in restricting the vesicular stomatitis virus [28,30]. However, pathogens can also take advantage of the functions or components of autophagy to enhance their replication and maintain intracellular survival, indicating that the relationship between autophagy and immunity is complicated and that different Atg proteins may have different roles [31].

Unlike Atg1, Atg5 or Atg8, the function of Atg2 is not well characterized. A previous study showed that Atg2 is related to lipid transport function [32]. In this study, we found that inhibiting *Atg2* induced a large number of melanotic nodules (Figure 1A–F). In fact, melanotic nodules have been studied for more than half a century and are regarded as a model of cancer [10]. The Toll, JAK/STAT, JNK and Ras/EGFR pathways participate in the formation of melanotic nodules [33,34,35]. The mechanism of melanotic nodule formation is complex, one of which involves the aggregation of hemocytes [36]. Consistent with this, plasmatocytes and lamellocytes were observed in melanotic nodules (Figure 1G–K). In addition, we observed more hemocytes in *Pxn* > *GFP/Atg2^RNAi^* larvae (Appendix A). However, lamellocytes are rare in healthy larvae, indicating that hematopoiesis is aberrant upon inhibition of *Atg2*. We will investigate how Atg2 controls hematopoiesis in further studies. Notably, compared to *Pxn* > *GFP/Atg2^RNAi^* larvae, when *Atg2* was knocked down using *Cg-Gal4*, a hemocyte- and fat-body-specific driver, more melanotic nodules were induced (Figure 1A–F), suggesting that an interaction between hemocytes and fat bodies may exist.

To examine whether the function of hemocytes was impaired, we used latex beads and pHrodo-conjugated pathogens to determine phagocytosis. pHrodo dye is sensitive to pH and will fluoresce in phagosomes whose pH is lower [37]. Based on this, we observed weaker fluorescent signals upon inhibition of *Atg2*, indicating that phagocytosis was suppressed (Figure 2). This finding is consistent with a previous study showing that Atg2 restricts the replication of *Mycobacterium marinum*, which involves the regulation of lipid droplets [38]. In our study, the proportion of NimC1 (P1)-positive hemocytes decreased upon inhibition of *Atg2* (Figure 3). In addition, we also showed that P1-positive hemocytes phagocytosed more latex beads than P1-negative hemocytes. Given that NimC1 is a marker for mature hemocytes and a typical phagocytosis receptor, this result suggested that the proportion of immature hemocytes increased, which led to disrupted in vivo phagocytosis. Different receptors are in charge of different types of pathogens. Whether Atg2 affects other receptors warrants further study. Given that the phagocytosis of nonpathogenic particle latex beads, the Gram-negative bacterium *E. coli* and Gram-positive bacterium *S. aureus* was disrupted, Atg2 played a broad role in controlling phagocytosis. Studies showed that filopodia act as “tentacles” to capture particles during phagocytosis [39,40]. In this study, the filopodia were altered by inhibiting *Atg2* (Figure 4). Similarly, Atg1 was shown to maintain F-actin protrusions of hemocytes, indicating that autophagy participates in actin cytoskeleton reorganization [41]. Moreover, our previous study revealed that Jumu regulates phagocytosis through NimC1 and cytoskeleton reorganization [42].

The secretion of AMPs by the fat body, equivalent to mammalian livers, is the hallmark of humoral immune responses of *Drosophila* [1,2]. This aspect of immunity is also known as the systemic immune response, which is efficient in eliminating invading pathogens. In addition, excessive AMP production damages cells [43]. In our study, the expression of AMP-encoding genes was altered by inhibiting *Atg2* (Figure 5); *Drs* was upregulated, and the transcript levels of *AttA, Dpt* and *Mtk* were decreased, indicating that the humoral immune response was disrupted. AMPs were shown to regulate aging and long-term memory beyond the role in mediating immune responses [2]. Moreover, in addition to the Toll and Imd pathways, JNK pathway transcription factor AP-1 also controls the expression of AMP-encoding genes [44]. These results suggested that both the role and the regulation of AMPs are complex. Thus, we speculated that Atg2 may regulate different AMPs in different manners. In line with the altered humoral immunity, flies were more susceptible to bacterial infections upon inhibition of *Atg2* (Figure 6). Although the contribution of cellular immune responses was questioned due to the prominent role of humoral immunity upon immune challenge, a previous study showed that plasmatocytes are required to promote AMP Def induction in the fat body during bacterial infections [45,46,47]. Consistent with this finding, knocking down *Atg2* only in hemocytes significantly decreased the survival rates after septic infection, suggesting that Atg2 may participate in the interactions between cellular and humoral responses.

Previous studies have shown that PGRP-LE activates autophagy in response to *L. monocytogenes*, indicating that PGRP-LE acts as an upstream regulator of autophagy [28]. However, in our study, Atg2 was upstream of the phagocytosis receptor and controlled its expression. Thus, the relationship between autophagy and the innate immune response is rather complex and requires further study. In conclusion, we have identified Atg2 as an important regulator of immune responses, which will advance the understanding of how autophagy functions in innate immunity.

## Figures and Tables

**Figure 1 insects-14-00706-f001:**
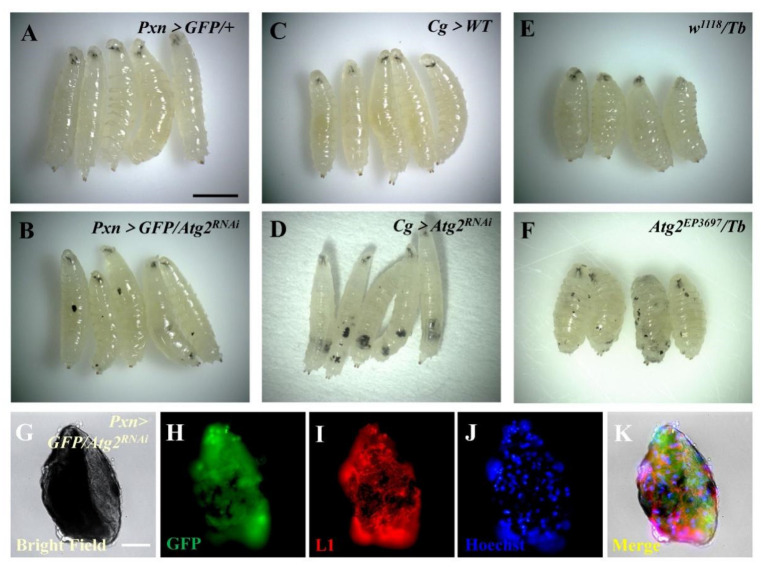
Inhibiting *Atg2* led to massive melanotic nodule formation. (**A**–**F**) The formation of melanotic nodules was observed in *Pxn* > *GFP/Atg2^RNAi^*, *Cg* > *Atg2^RNAi^* and *Atg2^EP3697^/Tb* larvae. (**G**–**K**) Immunostaining showed that plasmatocytes (GFP-positive cells, green) and lamellocytes (L1-positive cells, red) emerged in melanotic nodules from *Pxn* > *GFP/Atg2^RNAi^* larvae. Scale bars: 500 μm (larvae) and 50 μm (melanotic nodules).

**Figure 2 insects-14-00706-f002:**
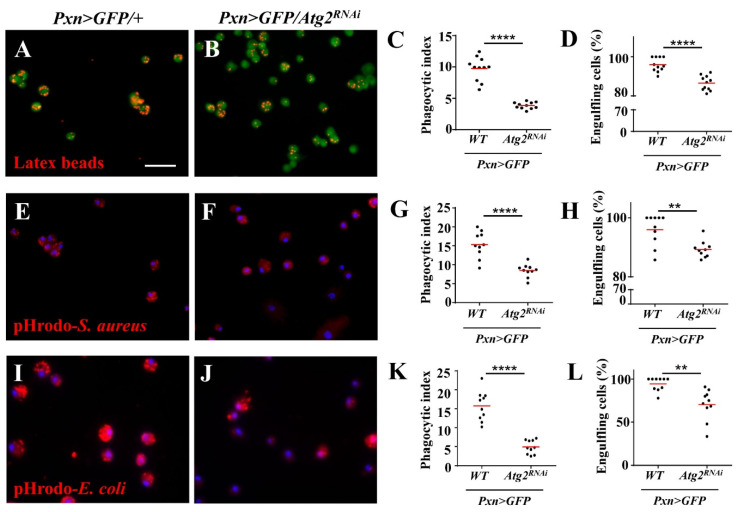
Knocking down *Atg2* in hemocytes disrupted the phagocytosis of latex beads and bacteria. (**A**–**L**) *Pxn* > *GFP/+* and *Pxn* > *GFP/Atg2^RNAi^* larvae were injected with latex beads (**A**,**B**), pHrodo-*S. aureus* (**E**,**F**) and pHrodo-*E. coli* (**I**,**J**). The quantification analysis showed that the average phagocytosed particles per hemocyte (phagocytic index) (**C**,**G**,**K**) and the percentage of engulfing hemocytes (hemocytes with phagocytosed particles) (**D**,**H**,**L**) significantly decreased when *Atg2* was knocked down in hemocytes. ** *p* < 0.01, **** *p* < 0.0001; scale bar: 25 μm.

**Figure 3 insects-14-00706-f003:**
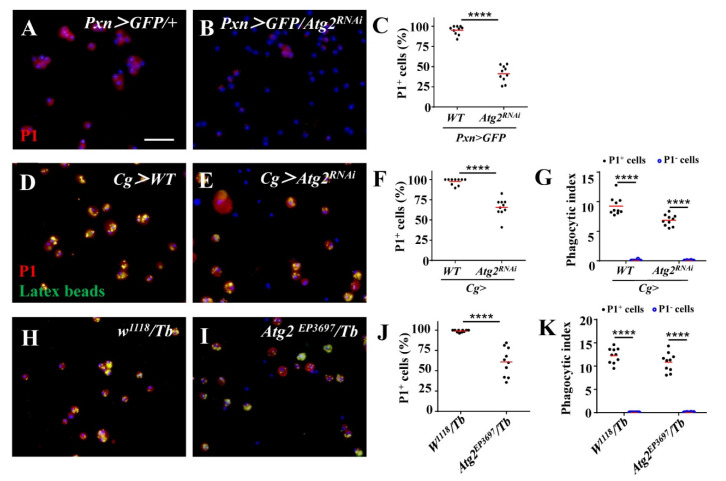
The proportion of NimC1-positive hemocytes decreased upon inhibition of *Atg2*. (**A**–**C**) The proportion of NimC1-positive hemocytes displayed by P1 staining significantly decreased in *Pxn* > *GFP/Atg2^RNAi^* hemocytes. (**D**–**K**) When injected with latex beads (green), the percentage of P1-positive hemocytes also decreased in *Cg* > *Atg2^RNAi^* and *Atg2^EP3697^/Tb* larvae. In addition, P1-positive hemocytes phagocytosed more latex beads than P1-negative hemocytes. **** *p* < 0.0001; scale bar: 25 μm.

**Figure 4 insects-14-00706-f004:**
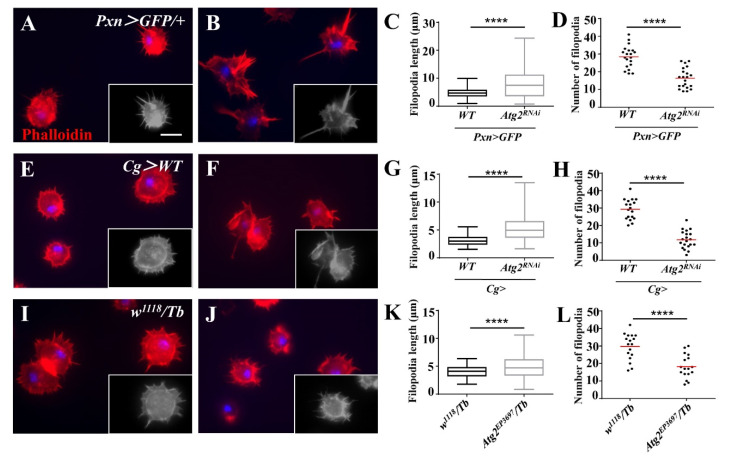
Inhibiting *Atg2* affected actin cytoskeleton distribution. (**A**–**L**) The hemocytes from *Pxn* > *GFP/+*, *Pxn* > *GFP/Atg2^RNAi^*, *Cg* > *WT*, *Cg* > *Atg2^RNAi^*, *w^1118^/Tb* and *Atg2^EP3697^/Tb* larvae were stained with phalloidin. The average filopodia length and filopodia number are shown in (**C**,**D**,**G**,**H**,**K**,**L**). To better identify filopodia, the grayscale images are also provided. The data showed that the hemocytes showed aberrant actin cytoskeleton distribution upon inhibition of *Atg2*. **** *p* < 0.0001; scale bar: 10 μm.

**Figure 5 insects-14-00706-f005:**
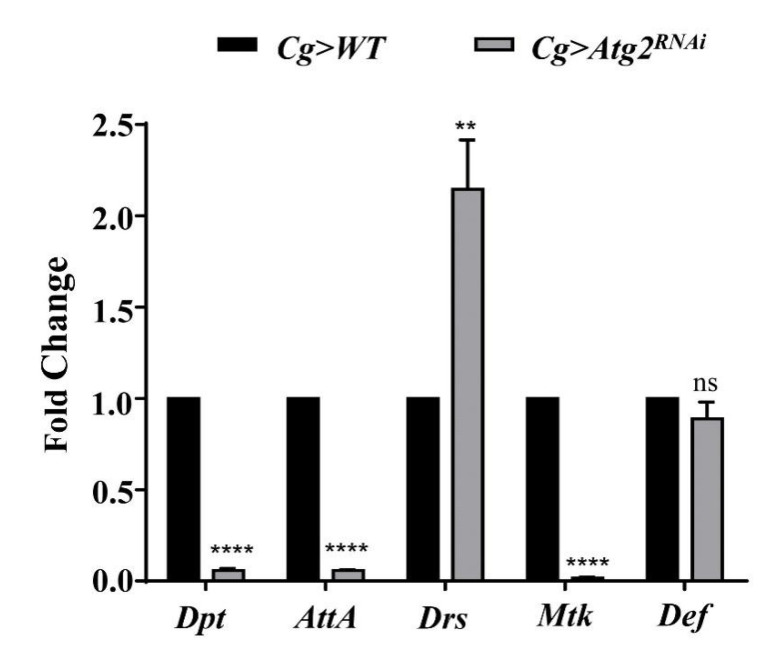
Inhibiting *Atg2* altered the expression of AMP-encoding genes. The transcript levels of AMP-encoding genes, including *Dpt*, *AttA*, *Drs*, *Mtk* and *Def* were determined by RT—qPCR, and the fold change was compared with the control group. The expression of *Drs* increased in *Cg* > *Atg2^RNAi^* larvae, while *Dpt*, *AttA* and *Mtk* were downregulated. ** *p* < 0.01, **** *p* < 0.0001, ns: not significant.

**Figure 6 insects-14-00706-f006:**
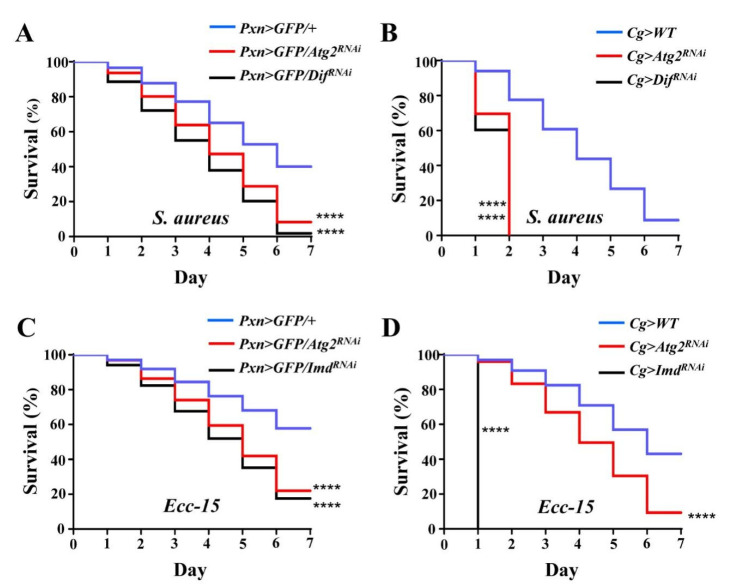
Flies are more susceptible to bacterial infection upon inhibition of *Atg2*. (**A**–**D**) Upon *S. aureus* and *Ecc-15* infections, *Pxn* > *GFP* > *Atg2^RNAi^* and *Cg* > *Atg2^RNAi^* flies showed significantly lower survival rates. *Pxn* > *GFP/Dif^RNAi^, Cg* > *Dif^RNAi^, Pxn* > *GFP/Imd^RNAi^* and *Cg* > *Imd^RNAi^* flies served as the positive controls and succumbed to *S. aureus* and *Ecc-15* infections. For each group, 15 males and 15 females were employed in the survival assay, and the experiment was repeated three times. The cumulative values of three replications are shown in the figures (**A**–**D**). **** *p* < 0.0001.

**Table 1 insects-14-00706-t001:** Primer sequences used qRT—PCR.

Target Gene	Forward (5’ to 3’)	Reverse (5’ to 3’)
*ribosomal protein 49*	AGTCGGATCGATATGCTAAGCTGT	TAACCGATGTTGGGCATCAGATACT
*Attacin-A*	AGGTTCCTTAACCTCCAATC	CATGACCAGCATTGTTGTAG
*Defensin*	CGCTTTTGCTCTGCTTGCTTGC	TAGGTCGCATGTGGCTCGCTTC
*Diptericin*	ATGCAGTTCACCATTGCCGTC	TCCAGCTCGGTTCTGAGTTG
*Drosomycin*	CTCTTCGCTGTCCTGATGCT	ATCCTTCGCACCAGCACTT
*Metchnikowin*	GCATCAATCAATTCCCGCCACC	CGGCCTCGTATCGAAAATGGG

## Data Availability

All data used in this paper are available within the text.

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
