# Peer review of "Atg2 Regulates Cellular and Humoral Immunity in Drosophila"

_insects, 2023, doi:10.3390/insects14080706_

Round 1
Reviewer 1 Report
Summary: In this manuscript, Qin et al report the involvement of the Atg2 gene in functioning of the Drosophila immune system. Atg2 has not been extensively studied in this regard and any new information is, therefore, a welcome addition. The authors demonstrate that various genetic knockdowns of Atg2 result in a notable underperformance of immune cells hemocytes, a lower expression of immune-active molecules, and overall reduced survivorship of infected larvae. The main novelty of this work is that it further strengthens a link between Atg2 and the immune system in addition to the initial suggestion published in 2017. The main criticism is that this study suffers from technical neglect and requires better justification of reported results.
Major points:
1. This study largely depends on quantified data, but the details of quantification analyses are not explicitly reported. That includes the numbers of analyzed cells and larvae, and how many cells have been averaged per larva. In addition, it is not clear what measures have been undertaken to ensure unbiased sample analysis by operators (e.g., had the samples been encrypted prior to analysis?).
2. Atg2 knockdown by Pxn driver causes hemocyte amplification, which may be a significant factor affecting the reported results (e.g. because of accumulation of immature hemocytes). Such effect of Pxn>Atg KD should be reported under the Results, included in the figure, and mentioned in the Discussion.
3. The claim that Atg KD affects NimC1 receptor expression requires additional validation using alternative methods, such as flow cytometry, Western, or qRT-PCR. In situ immunofluorescence alone is not adequate to support such a bold claim because it is not a quantitative method.
4. Changes in the actin cytoskeleton organization are overemphasized, but it’s mostly because of strong wording that was used. The authors are encouraged to provide a better evidence of ‘collapsed’ cytoskeleton, or tone down the discussion of their results reported in Fig. 4.
Minor
1. Add ID numbers to the stocks obtained from BDSC or VDSC.
2. Add catalogue numbers for commercial products used in the study (e.g. bacteria, latex particles, kits, etc)
3. What method was used to calculate qPCR results? Please add to the Methods with a reference.
4. Line 98: elaborate how bleeding was done; or was it squashing?
5. Figs 1, 2, 3: provide staining negative controls (with omitted primary antibody)
6. Fig.1: the quality of Fig. G-K is low and it does not clearly show the supposed co-localization of signals. The authors are encouraged to use confocal microscopy for best results.
7. Fig. 2: How was the scoring performed technically (e.g. sample encryption, multiples operators, etc.)? What does each dot in the graphs indicate? A cell? The average value from a single larva? Such details must be added to the Methods and figure legend. How do you ensure that all analyzed cells are hemocytes but not contaminating cells (for example, in fig S2 two types of nuclei are seen)?
8. Fig. 2: What measures have been undertaken by the researchers to ensure equivalent sample size of the samples that clearly have significantly different numbers of hemocytes (see S1 A and B)? Please address this issue in the Methods.
9. Fig. 2: How many cells have been analyzed? How many larvae have been analyzed? Add such data to all figures using quantification.
10. Fig. 2: the subpar quality of images in panels I and J undermines the value of the associated quantification – if the images used in quantification are difficult to interpret then the quantification itself becomes questionable. The authors are encouraged to select better images to showcase their point and take extra care not to over-compress the images (notably, the images from supplementary figures are of a better quality).
11. Fig. 4: to better demonstrate filopodia, the authors are encouraged to use grey scale instead of colored signal (red) that is less effectively detected by the human eye.
12. Fig. 6: The best practice requires infection trials to be repeated. How many flies have been used per experiment?
13. Correct to TUNEL (single “n”, all capital letters) in the axis of a graph in Fig. S1.
14. Line 291: please change “aggression” to “aggregation”
15. Figure 5: The increased expression of Drosomycin on the background of severely reduced expression of other antimicrobial peptides warrants some comments in the Discussion.
16. The authors are expected to provide better interpretation of the observed changes in filopodia. It is too premature to conclude that slightly shorter and fewer filopodia result in a “disrupted” actin cytoskeleton. Also, please provide a link connecting filopodia to phagocytosis as implied in the discussion.
Author Response
Major points:
- This study largely depends on quantified data, but the details of quantification analyses are not explicitly reported. That includes the numbers of analyzed cells and larvae, and how many cells have been averaged per larva. In addition, it is not clear what measures have been undertaken to ensure unbiased sample analysis by operators (e.g., had the samples been encrypted prior to analysis?).
We apologize for not providing details of quantification analyses. The details have been shown in the Method section. To avoid bias during the analysis, the samples were selected randomly and encrypted before they were analyzed.
- Atg2 knockdown by Pxn driver causes hemocyte amplification, which may be a significant factor affecting the reported results (e.g. because of accumulation of immature hemocytes). Such effect of Pxn>Atg KD should be reported under the Results, included in the figure, and mentioned in the Discussion.
We agree with the review that hemocyte count affects cellular immunity, and we did observe hemocyte amplication. In our study, we have injected excess of latex beads/pathogens, which should be enough to be phagocytosed by hemocytes. However, both the proportion of engulfing hemocytes and phagocytic index decreased in Atg2 KD group. We speculated that immature hemocytes accumulated as the reviewer pointed out, which led to disrupted in vivo phagocytosis. In addition, we did find that Atg2 KD significantly affected hematopoietic homeostasis, and this issue will be discussed in another manuscript that is currently under preparation. Thus, investigating how Atg2 affected hematopoiesis is out of the scope of this study.
- The claim that Atg KD affects NimC1 receptor expression requires additional validation using alternative methods, such as flow cytometry, Western, or qRT-PCR. In situ immunofluorescence alone is not adequate to support such a bold claim because it is not a quantitative method.
In this study, we showed that the proportion of NimC1 (P1)-positive hemocytes significantly decreased upon inhibition of Atg2. Given that P1 is a marker for mature plasmatocytes, this result suggested the proportion of immature hemocytes increased. In addition, these immature hemocytes may not have normal phagocytosis capacity like mature hemocytes. Consistent with this, we also showed that P1-positive hemocytes phagocytosed more latex beads than P1-negative hemocytes. We realized that the wording “expression of NimC1” is not appropriate, as we focused on the proportion of NimC1-positive cells. Thus, this wording has been modified in the manuscript.
- Changes in the actin cytoskeleton organization are overemphasized, but it’s mostly because of strong wording that was used. The authors are encouraged to provide a better evidence of ‘collapsed’ cytoskeleton, or tone down the discussion of their results reported in Fig. 4.
We agree that the wording about actin cytoskeleton organization is a sort of strong. Given that longer but few filopodia were observed in Atg2 KD group, we used “altered actin cytoskeleton” instead of “disrupted”.
Minor
- Add ID numbers to the stocks obtained from BDSC or VDSC.
The stock numbers have been provided.
- Add catalogue numbers for commercial products used in the study (e.g. bacteria, latex particles, kits, etc)
The catalogue numbers for commercial reagents have been provided.
- What method was used to calculate qPCR results? Please add to the Methods with a reference.
We calculated qPCR results with 2–∆∆Ct method. The detail and the reference were provided.
- Line 98: elaborate how bleeding was done; or was it squashing?
The bleeding method has been provided. To obtain circulating hemocytes, the larval cuticle was ripped gently with fine forceps while the larva was incubated with PBS. Then, the hemolymph will come out.
- Figs 1,2,3: provide staining negative controls (with omitted primary antibody)
We thank the reviewer for this suggestion. We have performed the staining without using the primary antibodies L1 and P1. As no staining was done in Figure 2, the negative controls for staining in Figure 1 and Figure 3 were provided. As shown in the figure below, although melanotic nodules were incubated with Alexa Fluor 568-conjugated secondary antibodies, no fluorescence signals were seen without using anti-L1 antibodies. Similar to this, no P1 signals were observed in hemocytes from larvae with or without being injected with latex beads. These data indicated that the red fluorescence signals in our figures were not non-specific.

- 1: the quality of Fig. G-K is low and it does not clearly show the supposed co-localization of signals. The authors are encouraged to use confocal microscopy for best results.
Unfortunately, our confocal microscope is currently under maintenance. To obtain figures of a better quality, we reperformed the staining of melanotic nodules, and observed them under a fluorescence microscope.
- Fig. 2: How was the scoring performed technically (e.g. sample encryption, multiples operators, etc.)? What does each dot in the graphs indicate? A cell? The average value from a single larva? Such details must be added to the Methods and figure legend. How do you ensure that all analyzed cells are hemocytes but not contaminating cells (for example, in fig S2 two types of nuclei are seen)?
The details have been added to the Method section. To avoid individual bias, sample encryption was employed in the analysis. In the graph, each dot indicates the average phagocytosed particles per hemocyte. In fact, it is difficult for other cell types to attach to slides upon so many times of washes with PBST in our study. Given that two types of nuclei were seen in Figure S2, we speculated that the larger nuclei were from lamellocytes.
- Fig. 2: What measures have been undertaken by the researchers to ensure equivalent sample size of the samples that clearly have significantly different numbers of hemocytes (see S1 A and B)? Please address this issue in the Methods.
Before performing the experiment, the larvae were staged by using 0.4% Trypan Blue. To ensure equivalent sample size, same amount of larvae with same developmental stage were used for each group.
- Fig. 2: How many cells have been analyzed? How many larvae have been analyzed? Add such data to all figures using quantification.
The data have been provided in the Method section.
- Fig. 2: the subpar quality of images in panels I and J undermines the value of the associated quantification – if the images used in quantification are difficult to interpret then the quantification itself becomes questionable. The authors are encouraged to select better images to showcase their point and take extra care not to over-compress the images (notably, the images from supplementary figures are of a better quality).
We have changed figures in order to better display our results. In addition, the figures are without over-compression.
- Fig. 4: to better demonstrate filopodia, the authors are encouraged to use grey scale instead of colored signal (red) that is less effectively detected by the human eye.
To help readers better identify filopodia, we have shown the grey version in Figure 4. In fact, in our study, the figures were first converted into 8-bit images and then analyzed using ImageJ.
- 6: The best practice requires infection trials to be repeated. How many flies have been used per experiment?
For each group, 15 males and 15 females were employed, and the infection was repeated for 3 times. Thus totally 90 flies were used for each group. The details have been provided.
- Correct to TUNEL (single “n”, all capital letters) in the axis of a graph in Fig. S1.
The axis has been revised.
- Line 291: please change “aggression” to “aggregation”
We apologize for this typo. The “aggression” was revised to “aggregation”.
- Figure 5: The increased expression of Drosomycin on the background of severely reduced expression of other antimicrobial peptides warrants some comments in the Discussion.
We thank the reviewer for this suggestion, and this issue has been discussed in the Discussion section.
- The authors are expected to provide better interpretation of the observed changes in filopodia. It is too premature to conclude that slightly shorter and fewer filopodia result in a “disrupted” actin cytoskeleton. Also, please provide a link connecting filopodia to phagocytosis as implied in the discussion.
We realized that the wording “disrupted” is too strong. Given that longer but fewer filopodia were seen in the experimental group, we used “altered actin cytoskeleton” instead. A link connecting filopodia to phagocytosis has been added to the Discussion section.
Reviewer 2 Report
Manuscript insects-2506658 entitled “Atg2 regulates cellular and humoral immunity in Drosophila by Bo Qi et al. represents a straightforward study maximally exploiting the advantages of the availability of Drosophila genetic strain crossings that results in Atg2 knockdown/knockout. The experiments are all well designed, include correct controls and are supported by correct statistical power. The authors recognize that although their experiments underline the involvement of Atg2 in nodule formation, phagocytotic activity by hemocytes and differential expression of AMPs, the overall role of Atg2 is complex as many additional interactions need to be studied. Anyway an important outcome of this study is the observation that hampered Atg2 expression makes these flies less resistant to pathogen challenges and as such might provide a nice target for insect control.
In future experiments the authors should pay more attention to changes in overall as well as differential hemocytes counts following Atg2 knockdown. Indeed the strongly enhanced nodule formation in Atg2 knockdown animals calls for hemocytes replenishment. In the present study the authors also ignore the direct involvement of hemocytes in humoral immunity as most, if not all, AMPs are synthesized by both fatbody (this study) and hemocytes (see literature).
Paragraph 3.2 suggests phagocytosis of hemocytes whereas the authors mean phagocytosis of beads/pathogens by the hemocytes
Author Response
In future experiments the authors should pay more attention to changes in overall as well as differential hemocytes counts following Atg2 knockdown. Indeed the strongly enhanced nodule formation in Atg2 knockdown animals calls for hemocytes replenishment. In the present study the authors also ignore the direct involvement of hemocytes in humoral immunity as most, if not all, AMPs are synthesized by both fatbody (this study) and hemocytes (see literature). Paragraph 3.2 suggests phagocytosis of hemocytes whereas the authors mean phagocytosis of beads/pathogens by the hemocytes
We agree with the reviewer that increased hemocyte count is associated with melanotic nodule formation. In fact, we did notice that inhibiting Atg2 significantly disrupted hematopoietic homeostasis, and the detailed mechanism will be discussed in another manuscript, which is under preparation. We also agree that hemocytes themselves can secret AMPs, and we will address this issue in our future studies. Finally, we would like to thank the reviewer for the correction. We realized that the wording “phagocytosis of hemocytes” is confusing, and the “of hemocytes” was deleted in the manuscript.

Round 2
Reviewer 1 Report
Reviewer is pleased with authors’ responses and manuscript edits. Below are the minor things that still need attention.
1. Include “unpublished images” containing immunofluorescence negative controls as supplementary information;
2. Line 117: add how many larvae have been used in hemocyte counts per condition;
3. Legend of Figure 4: Include explanation of monochrome insets added to the figure;
4. Legend of Figure 6: clarify what is shown in the graphs (e.g. the average of repetitive trials, cumulative values combined from three trails, or results of the best trial);
5. Line 330: please italicize "in vivo";
6. Line 339: the last sentence on this line can be omitted as trivial.
Author Response
1. Include “unpublished images” containing immunofluorescence negative controls as supplementary information;
2. Line 117: add how many larvae have been used in hemocyte counts per condition;
3. Legend of Figure 4: Include explanation of monochrome insets added to the figure;
4. Legend of Figure 6: clarify what is shown in the graphs (e.g. the average of repetitive trials, cumulative values combined from three trails, or results of the best trial);
5. Line 330: please italicize "in vivo";
6. Line 339: the last sentence on this line can be omitted as trivial.
We thank the reviewer for the suggestions, and these issues have been addressed.
Round 3
Reviewer 1 Report
Thank you and best of luck to you.
Author Response
On behalf of my co-authors, we thank you very much for handling our manuscript, we appreciate you very much for your positive and constructive comments and suggestions on our manuscript.